# Cytomegalovirus reactivation in mechanically ventilated patients with or without SARS-CoV-2 infection: A retrospective cohort study

Octave Cannac[1], Vanessa Pauly[2], Christine Zandotti[3,4], Léa Luciani[3], Paul-Rémi Petit[3,5], Xavier de Lamballerie[3], Rémi Charrel[3,6,5], Laurent Papazian[7,8], Damien Barrau[1], Geoffray Agard[1,7], Sami Hraiech[1,7]*

1 Service de Médecine Intensive Réanimation, APHM, Hôpital Nord, Marseille, France, 2 Aix Marseille University, School of Medicine - La Timone Medical Campus, EA 3279, CEReSS - Health Services Research and Quality of life Center, Marseille, France, 3 Unité des Virus Émergents (UVE), Aix-Marseille Univ-IRD 190-Inserm 1207, Marseille, France, 4 Laboratoire BVH –Virologie – Aigue – Tropique- AP-HM, Marseille, France, 5 Le Service de Prévention du Risque Infectieux (LESPRI), CLIN AP-HM Hôpitaux Universitaires de Marseille, Marseille, France, 6 Laboratoire des Infections Virales Aigues et Tropicales, Pole des Maladies Infectieuses, AP-HM Hôpitaux Universitaires de Marseille, Marseille, France, 7 Aix-Marseille Université, Faculté de Médecine, Centre d'Études et de Recherches sur les Services de Santé et qualité de vie EA 3279, Marseille, France, 8 Service de Réanimation, Intensive Care Unit, Centre Hospitalier de Bastia, Bastia, France

* sami.hraiech@p-hm.fr

## Abstract

### Background

Data comparing the incidence, risk factors and outcomes of cytomegalovirus (CMV) reactivation in SARS-CoV-2 positive and negative patients remain controversial.

### Design

A retrospective cohort study in a tertiary center.

### Patients

Patients admitted to the intensive care unit between December 2019 and May 2021, under invasive mechanical ventilation for 4 days or more and screened for CMV reactivation were included.

### Measurements

The primary outcome was the incidence of CMV reactivation in SARS-CoV-2 negative and positive patients. Secondary outcomes included risk factors for CMV reactivation in both populations and survival analysis according to CMV reactivation in SARS-CoV-2 negative and positive patients.

**Data availability statement:** All relevant data are within the paper and its Supporting information files.

**Funding:** The author(s) received no specific funding for this work.

**Competing interests:** The authors have declared that no competing interests exist.

## Results

CMV reactivation occurred in 34.7% (n = 51/147) of SARS-CoV-2 negative patients and in 45.4% (83/183) of SARS-CoV-2 positive patients ($p = 0.08$). When considering competing factors, SARS-CoV-2 infection was not associated with CMV reactivation (sub-hazard ratio (SHR) = 1.01 [0.68–1.49], $p = 0.98$). Treatment with methylprednisolone was significantly associated with CMV reactivation in the unadjusted and adjusted analysis (SHR 2.81 [2.01–3.93] $p < 0.001$; SHR 2.84 [1.94–4.15] $p < 0.001$ respectively). Patients combining SARS-CoV-2 infection and CMV reactivation had a significantly higher all-cause mortality. Among patients presenting a CMV reactivation, the administration of ganciclovir was a protective factor for day-60 mortality (HR = 0.4; [0.22–0.74] $p = 0.004$).

## Conclusion

In this large retrospective cohort, CMV reactivation was not significantly associated with SARS-CoV-2 infection in patients undergoing invasive mechanical ventilation for at least 4 days. The major risk factor for CMV reactivation was treatment with methylprednisolone. The combination of CMV reactivation with SARS-CoV-2 infection was associated with a higher mortality whereas ganciclovir treatment reduced mortality.

## Background

The rate of Cytomegalovirus (CMV) reactivation among immunocompetent intensive care unit (ICU) patients has been reported to range from 8% to 46% [1]. After primary-infection, CMV stays in a latent state in targeted human cells but it is able to evade host immune system especially during severe infections [2]. Several studies have highlighted the association between CMV reactivation and increased mortality, length of invasive mechanical ventilation (IMV) and length of stay in ICU patients [3]. The shift to an active, aggressive and morbid viral infection often occurs when the host's immune system suffers either a repression or an overactivation even in previously immunocompetent patients, such as during septic shock [4,5]. The hypotheses behind the possible damage caused by CMV in critically ill patients are its immunomodulation capacities resulting in more bacterial and fungal co-infections; its direct cytopathic effect, especially on pneumocytes; and the acceleration of lung fibrosis resulting in persistent lung injury [6–8].

In severe forms of infection by the severe acute respiratory syndrome coronavirus-2 (SARS-CoV-2), patients present an acute respiratory distress syndrome (ARDS) often in relation with a major cytokine storm in reaction to the viral infection [9]. This leads to high mortality and prolonged IMV. In vitro data suggest that CMV reactivation in epithelial cells facilitates SARS-CoV-2 superinfection by upregulating a key viral surface receptor suggesting a pathophysiological explanation for poorer prognosis in co-infected patients [10].

CMV reactivation in SARS-CoV-2 critically patients has been assessed in several studies, however often in limited effective cohorts with heterogeneous design [10–17]. Conflicting results exist concerning the incidence of CMV reactivation in SARS-CoV-2 patients and it is unclear whether SARS-CoV-2 secondary triggers CMV reactivation. Moreover, a negative effect of CMV in SARS-CoV-2 patients with longer duration of IMV and increased mortality has been inconsistently described [11–13].

We conducted a retrospective cohort study evaluating CMV reactivation in SARS-CoV-2 positive and negative patients. We aimed to determine whether CMV reactivates more frequently in SARS-CoV-2 positive patients. Our secondary objectives were to evaluate the risk factors associated with CMV reactivation and if it affects differently the prognosis of SARS-COV-2 positive and negative patients.

## Materials and methods

### Study design and population

In this retrospective observational study, we included all patients from December 1st, 2019 to May the 31st, 2021, who were admitted to the Medical Intensive Care Unit of the University Hospital of Marseille, France, and who underwent at least 4 days of IMV. Patients were excluded if they were under 18 years of age and if no test for CMV in blood or lower respiratory tract (LRT) were performed during their stay.

### Ethics

Considering the retrospective and observational design of the study, according to French Law (Loi Jardé, méthodologie de reference MR004), consent was not required. Patients and/or their relatives were however informed of the anonymous data collection and given the opportunity to decline inclusion. This information was delivered through an information booklet given to the patients and their relatives. The Ethics Committee of the French Intensive Care Society (Société de Réanimation de Langue Française) approved the study protocol (approval number CE SRLF 24−017). The study was declared to the University Hospital Legal Authority (Registre Général de Protection des Données) and registered under the number PADS24–21_dgr.

### Data collection

All data were collected retrospectively using the patients' electronic files which were accessed from May 2nd 2024 to May 31st 2024 and collected on an anonymized data base not containing information that could identify individual participants.

Patient's demographics, medical history, Sequential Organ Failure Assessment (SOFA), Simplified Acute Physiology Score (SAPS) II, CMV serological status when available, CMV reactivation status, SARS-CoV-2 status were collected. The use of immunosuppressive or immunomodulatory drugs such as corticosteroids (methylprednisolone and dexamethasone separately registered) and anti-cytokine agents (anti-IL1, anti-IL6 and anti-Janus kinase) in SARS-CoV-2 patients was recorded. Anti-CMV curative therapy (ganciclovir) was noted. The need for renal replacement therapy (RRT), veno-venous extra corporeal membrane oxygenation (ECMO), duration of IMV, episodes of bacterial ventilator associated pneumonia (VAP), date of death were also collected.

### CMV reactivation and SARS-CoV-2 positivity

During the study period, our ICU routine care protocol planned that patients under IMV for at least 4 days were screened twice weekly for CMV reactivation with polymerase chain reaction (PCR) on whole blood for as long as they remained under IMV. This had become part of routine care since a randomized controlled trial [14] investigating the effect of pre-emptive ganciclovir in ICU patients with CMV reactivation. LRT CMV PCR were performed according to clinicians' decision. Patients with at least one Polymerase Chain Reaction (PCR) test (R-GENE® for LRT samples, NeuMoDx™ for blood samples) finding CMV DNA above 50 UI/mL in a blood sample or above 500 UI/mL in a LRT sample were considered in

the CMV reactivation group. SARS-CoV-2 diagnosis was made by positive RT-PCR on either LRT sample or nasal swab (Real time fluorescent RT-PCR kit for detecting 2019-nCov Ref: MFG030010 (MGI)).

## Data analysis

First, we compared patients' demographics, medical history, treatments and outcomes for SARS-CoV-2 positive and negative patients. For quantitative variables, Student's t-test was performed when following a parametric distribution. Otherwise, the Mann-Whitney rank sum test was performed. Concerning qualitative or binary variables, Chi-squared or Fisher's exact test were used to compare percentages.

Then, to analyze if SARS-CoV-2 was associated with the incidence of CMV reactivation within 60 days following intubation, we performed competing risk analysis with death and extubation (before CMV reactivation) as the competing events. We used the Fine and Gray model to estimate the marginal probability of 60-days CMV reactivation in the presence of competing events (*i.e.*, death and extubation) and to directly model the effect of covariates on the Cumulative Incidence Function (CIF) for CMV reactivation. We provided stacked plot of events (CMV reactivation, death, extubation, censored) for the 2 groups which are cumulative incidence functions for each event type estimated using the Aalen-Johansen method to account for competing risks. These functions represent the estimated probability over time of experiencing each event, treating other event types as competing events,

To analyse overall survival (truncated at 60 days post intubation), we performed a survival analysis using the Cox regression model.

For both primary (60-days incidence of CMV reactivation) and secondary (60-days mortality) outcomes, we performed two distinct multivariable analyses adjusting using the backward selection strategy of variables: for each of the two endpoints, we first performed unadjusted analyses on distinct factors (patients' demographics, medical history, treatments and outcomes) and selected factors that were associated with a p-value<0.20 to enter into the multivariate model. Then, the backward selection into the multivariate models consisted on keeping only variables that were associated with a p-value<0.05. Start date corresponded to the day of the intubation, data were censored at 60 days following the start date. The SARS-CoV-2 positive variable was forced into the multivariate analysis: as the main factor of analysis, this variable was included in the multivariable model regardless of its statistical significance in univariate screening. CMV reactivation was considered as a time dependent variable in the mortality analysis (the covariate status for each individual changes from 0 to 1 at the time of CMV activation, i.e., individuals were considered unexposed until the time of CMV activation and exposed thereafter).

For the secondary outcomes, we performed a complementary analysis while replacing the two variables CMV reactivation and SARS-CoV-2 by their combination (CMV reactivation in SARS-CoV-2 patients, CMV reactivation in non SARS-CoV-2 patients, no CMV reactivation in SARS-CoV-2 patients, no CMV reactivation in non SARS-CoV-2 patients) in order to distinguish between clinical groups and be more informative.

We finally performed a subgroup analysis of factors associated with mortality only for patients who reactivated CMV.

The previous sensitivity analyses were repeated in the sub-population of patients with a known seropositive CMV status at ICU admission.

Results were expressed as Sub-Hazard ratios (SHR) and their 95% confidence intervals for the primary endpoint considering CMV reactivation (Fine and Gray model) and with Hazard ratios (HR) and their 95% confidence intervals for the secondary endpoint (Cox model) considering mortality. We provided plots of cumulative incidence of reactivation as well as survival curves.

All statistical analyses were performed with the SAS software®version 9.4. (SAS Institute Inc, SAS 9.4, Cary, NC).

## Results

### Patients' characteristics

During the study period, 1036 patients were admitted to the recruiting ICU. Among them, 662 underwent IMV. Three hundred and thirty patients were ventilated for 4 days or more and had at least one test for CMV reactivation in either blood or

LRT samples (Fig 1). Patients' main characteristics are summarized in Table 1. CMV serological status at ICU admission was available for 166 (50.3%) patients with a comparable repartition in SARS-CoV-2 positive (91/183; 49.7%) and negative (75/147; 51.0%) groups. Among patients with a known CMV serological status, 77.8% were seropositive (78% for SARS-CoV-2 positive and 77.3% for SARS-CoV-2 negative patients; p = 0.95).

### Incidence of CMV reactivation among SARS-CoV-2 positive and negative patients

A hundred and thirty-eight patients (41.8%) of the overall population presented CMV reactivation.

SARS-CoV-2 negative patients presented less CMV reactivation 34.7% (n = 51/147) than SARS-CoV-2 positive patients (45.4% (83/183)) but the latter experienced less frequently competing events (42.1% for death or 10.4% for extubation) vs. (46.3% for death and 16.3% for extubation) for SARS-CoV-2 negative patients (Fig 2).

Neither the univariate nor multivariate analyses yielded significant associations between SARS-CoV-2 and CMV reactivation (multivariate (SHR) = 1.01 [0.68–1.49], p = 0.98) (Table 2).

### Risk factors for CMV reactivation among SARS-CoV-2 positive and negative patients

Treatment with methylprednisolone was significantly associated with CMV reactivation in the unadjusted and adjusted analyses (SHR 2.81 [2.01–3.93], p < 0.001; SHR 2.84 [1.94–4.15], p < 0.001 respectively); Table 2). When tested in our model, interaction between treatment with methylprednisolone and SARS-CoV-2 status was not significant (p = 0.91) suggesting that the effect of methylprednisolone on CMV reactivation did not differ according to SARS-CoV-2 status. Treatment with dexamethasone was not associated with CMV reactivation (p = 0.23) in the overall population.

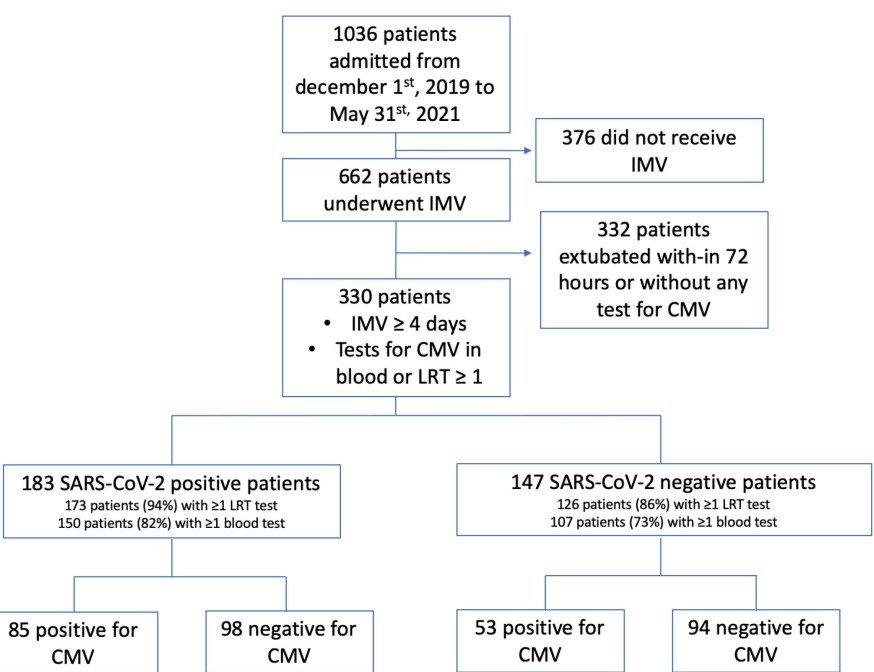

**Fig 1. Flow chart of the study.** *IMV: Invasive Mechanical Ventilation; CMV: Cytomegalovirus; LRT: Lower Respiratory Tract, SARS-CoV-2: Severe acute respiratory syndrome coronavirus 2.*

**Table 1. Patients' baseline characteristics according to their SARS-CoV-2 status in the overall population.** *p-values for unadjusted analysis.*

| | Overall population N=330 | SARS-CoV-2 positive patients N=183 | SARS-CoV-2 negative patients N=147 | p value |
|---|---|---|---|---|
| Age, years *mean±SD* | 60±13 | 61±11 | 58±16 | 0.07 |
| Male *n(%)* | 231 (70) | 139 (76) | 92 (62.6) | 0.01 |
| BMI, kg/m², *mean±SD* | 28±6 | 30±5 | 26±6 | <0.001 |
| Cardiovascular diseases *n(%)* | 197 (59.7) | 110 (60.1) | 87(59.2) | 0.91 |
| Diabetes mellitus *n(%)* | 89 (27) | 58 (31.7) | 31 (21.1) | 0.03 |
| Current smoker *n(%)* | 126 (38.2) | 50 (27.3) | 76 (51.7) | <0.001 |
| Alcohol consumption *n(%)* | 21 (6.4) | 4 (2.2) | 17 (11.6) | <0.01 |
| Chronic lung disease *n(%)* | 110 (33.3) | 34 (18.6) | 76 (51.7) | <0.001 |
| Chronic renal failure *n(%)* | 19 (5.8) | 13 (7.1) | 6 (4.1) | 0.34 |
| Solid cancer *n(%)* | 53 (16.1) | 12 (6.6) | 41 (27.9) | <0.001 |
| Hematologic malignancy *n(%)* | 19 (5.8) | 5 (2.7) | 14 (9.5) | 0.02 |
| Immunocompromised status *n(%)* | 57 (17.3) | 17 (9.3) | 40 (27.2) | <0.001 |
| Known CMV serological status | 166 (50.3) | 91 (49.7) | 75 (51.0) | 0.82 |
| CMV seropositive status*, n (%) | 129 (77.8) | 71 (78) | 58 (77.3) | 0.95 |
| **Severity scores at ICU admission** *median(IQR)* | | | | |
| SAPS2 score | 44.5 (36-44.5) | 41 (34-49) | 50 (40.50-60) | <0.001 |
| SOFA | 8 (4-8) | 7 (4-8) | 8 (6-11) | <0.001 |
| **Main reasons for ICU admission** *n(%)* | | | | |
| Acute respiratory failure | 262 (79.4) | 171 (93.4) | 91 (61.9) | <0.001 |
| Septic shock | 27 (8.2) | 2 (1.1) | 25 (17) | <0.001 |
| Post-operative | 38 (11.5) | 1 (0.5) | 37 (25.2) | <0.001 |
| Neurological disorder | 15 (4.5) | 1 (0.5) | 14 (9.5) | <0.001 |
| Cardiogenic shock | 4 (3.2) | 1 (0.5) | 3 (2) | 0.33 |
| Cardiac arrest | 9 (2.7) | 1 (0.5) | 8 (5.4) | 0.01 |

SAPS2: Simplified Acute Physiology Score 2; SOFA: Sequential Organ Failure Assessment, ICU: intensive Care Unit; SD: Standard Deviation; IQR: Interquartile Range.

*Among the patients with known serological status.

## Analyses of 60-day mortality

CMV reactivation was found to be significantly associated with all-cause mortality in the 60 days following intubation (HR 1.70 [1.12–2.57], *p*=0.01; Table 3).

Patients combining SARS-CoV-2 infection and CMV reactivation had a significantly higher all-cause mortality compared to SARS-CoV-2 negative patients with CMV reactivation or patients with CMV reactivation no matter their SARS-CoV-2 status (Fig 3, Table 4).

The use of curative antiviral treatment against CMV in patients with reactivation was found to be a protective factor for day 60 mortality after adjustment on SARS-CoV-2 status (HR=0.4; [0.22–0.74] *p*=0.004) (Table 5).

## Analyses in the sub-population with known seropositive CMV status

Sensitivity analyses performed in patients with a known and positive CMV serological status at ICU admission are reported in the supplementary data (S1 and S2 Tables). The use of methylprednisolone remained significantly associated with CMV reactivation in the unadjusted and adjusted analyses (SHR 2.73 [1.71–4.35], p<0.001 and SHR 2.76[1.53–4.96], p<0,001 respectively).

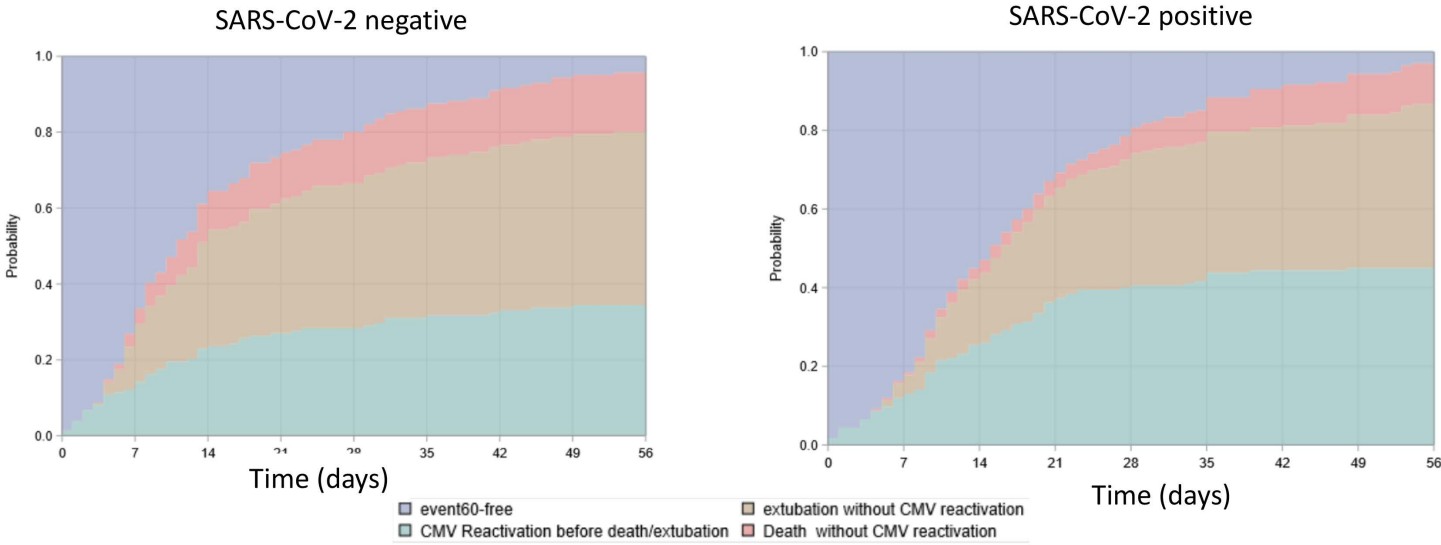

Fig 2. Stacked plots representing cumulative incidence of the different events (CMV reactivation, extubation without CMV reactivation, death without CMV reactivation, censoring) in the first 60 days post intubation according to SARS-CoV-2 status. *CMV = Cytomegalovirus.*

Table 2. Factors associated with CMV reactivation in overall population. Stepwise downward selection method forcing the variable « SARS-CoV-2 positive ».

| | Unadjusted | | | Adjusted | | |
|---|---|---|---|---|---|---|
| | SHR | CI at 95% | P-value | SHR | CI at 95% | P-value |
| SARS-CoV-2 positive | 1.37 | 0.97-1.94 | 0.08 | 1.01 | 0.68-1.49 | 0.98 |
| Methylprednisolone | 2.81 | 2.01-3.93 | <0.001 | 2.84 | 1.94-4.15 | <0.001 |
| Time from hospital admission to intubation | 1.03 | 1.01-1.05 | <0.01 | 1.03 | 1.01-1.05 | <0.01 |
| Dexamethasone | 1.23 | 0.88-1.72 | 0.23 | | | |
| ARDS | 2.31 | 1.19-4.49 | 0.01 | | | |
| ECMO | 1.70 | 1.22-2.38 | <0.01 | | | |
| Blood transfusion | 1.97 | 1.34-2.89 | <0.001 | | | |
| Prone positioning | 1.59 | 1.06-2.39 | 0.02 | | | |
| SOFA | 0.96 | 0.92-1.01 | 0.13 | | | |
| VAP occurence | 1.34 | 0.95-1.88 | 0.09 | | | |

CMV = Cytomegalovirus; SHR = Sub-Hazard Ratio; CI = Confidence Interval; RR = Relative Risk, ARDS = Acute Respiratory Distress Syndrome, ECMO = Extra Corporeal Membrane Oxygenation, SOFA = Sequential Organ Failure Assessment, VAP = Ventilator Associated Pneumonia.

## Discussion

In this large retrospective cohort, we found that CMV reactivation was not significantly associated with SARS-CoV-2 infection in patients undergoing IMV for at least 4 days. The major risk factor for CMV reactivation was treatment with methylprednisolone but not dexamethasone. The combination of CMV reactivation in patients under IMV with SARS-CoV-2 infection was associated with a higher mortality at day-60 and the administration of ganciclovir was a protective factor towards mortality.

Our series is one of the largest comparative retrospective study to date on CMV reactivation in critically ill SARS-CoV-2 positive patients, confirming the higher mortality of combined viral infections. It is the first to distinguish the role of

**Table 3. Risk factors for mortality in the first 60 days post intubation in the overall population, adjusted and unadjusted analysis.**

| | Unadjusted | | | Adjusted | | |
|---|---|---|---|---|---|---|
| | HR | CI at 95% | P-value | HR | CI at 95% | P-value |
| SARS-CoV-2 | 0.89 | 0.61-1.30 | 0.54 | 1.87 | 1.14-3.07 | 0.01 |
| CMV reactivation | 1.85 | 1.24-2.76 | <0.01 | 1.70 | 1.12-2.57 | 0.01 |
| Cancer | 2.69 | 1.74-4.14 | <0.001 | 2.26 | 1.36-3.75 | 0.002 |
| Alcohol consumption | 2.00 | 1.07-3.74 | 0.03 | 2.13 | 1.10-4.13 | 0.03 |
| Sepsis | 2.58 | 1.52-4.39 | <0.001 | 1.99 | 1.07-3.69 | 0.03 |
| VAP occurence | 0.50 | 0.434-0.73 | <0.001 | 0.52 | 0.35-0.77 | <0.01 |
| Age | 1.05 | 1.03-1.07 | <0.001 | 1.03 | 1.01-1.05 | <0.01 |
| SAPS 2 score | 1.03 | 1.02-1.04 | <0.001 | 1.03 | 1.01-1.04 | <0.001 |
| Time from hospital admission to intubation | 1.02 | 1.01-1.04 | 0.01 | 1.02 | 1.00-1.04 | 0.05 |

VAP = Ventilator Associated Pneumonia; SAPS 2 = Simplified Acute Physiology Score 2; CMV = Cytomegalovirus; HR = Hazard Ratio; CI = Confidence Interval.

NB: CMV reactivation was treated as a time dependent variable.

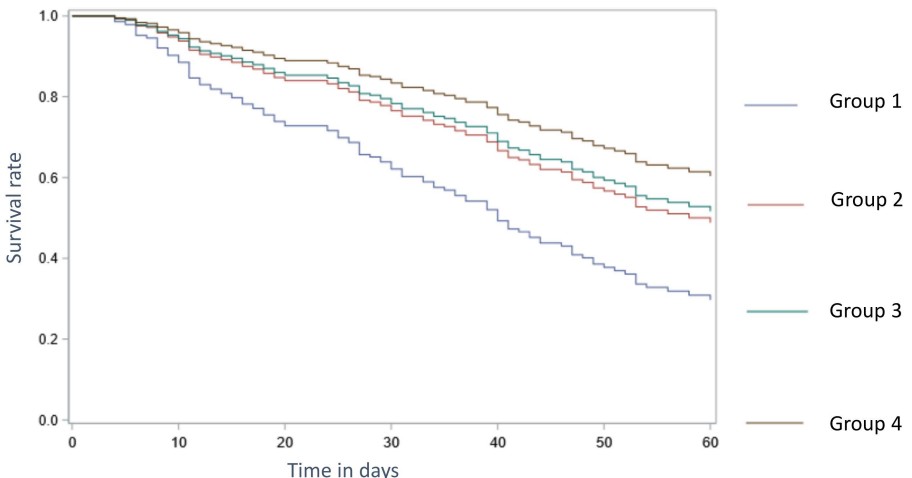

**Fig 3. Adjusted survival by group of patients (Cox model adjusted on covariables listed in table 3).** *Group 1 = SARS-CoV-2 positive patients with CMV reactivation. Group 2 = SARS-CoV-2 positive patients without CMV reactivation. Group 3 = SARS-CoV-2 negative patients with CMV reactivation. Group 4 = SARS-CoV-2 negative patients without CMV reactivation.*

methylprednisolone as a risk factor for CMV reactivation and to highlight the protective effect of treatment with ganciclovir in this population.

Our study showed an incidence of CMV reactivation in either blood or LRT samples of 45.4% in mechanically ventilated SARS-CoV-2 positive patients, comparably to what was described elsewhere [13,15,16]. One of the originality of our work is that we screened not only blood but also respiratory tract reactivation. Rates of CMV reactivation among ICU SARS-CoV-2 positive patients range from 0% to 47% depending on how viral reactivation was defined and patients' severity [11,17–21]. Boers et al. studied one of the largest retrospective monocentric cohort of SARS-CoV-2 positive patients with IgG directed against CMV, undergoing mechanical ventilation [21]. They found 5.7% of CMV reactivation defined as more than $10^4$ copies/mL in a LRT sample which is significantly lower than our results. This difference can be explained by the unusually high threshold chosen to define a clinically relevant CMV reactivation as well as not considering blood

**Table 4. Comparison of survival at 60 days post intubation depending on CMV reactivation and SARS-CoV-2 status.**

| Groups compared | Hazard Ratio | CI at 95% |
|---|---|---|
| 1 vs 2 | **2.006** | **1.14-3.54** |
| 1 vs 3 | **2.230** | **1.18-4.23** |
| 1 vs 4 | **3.129** | **1.76-5.57** |
| 2 vs 3 | 1.112 | 0.54-2.30 |
| 2 vs 4 | 1.560 | 0.82-2.98 |
| 3 vs 4 | 1.403 | 0.77-2.57 |

After adjustment on confounding factors described in Table 3.

CI = Confidence Interval.

Group 1 = SARS-CoV-2 positive patients with CMV reactivation.

Group 2 = SARS-CoV-2 positive patients without CMV reactivation.

Group 3 = SARS-CoV-2 negative patients with CMV reactivation.

Group 4 = SARS-CoV-2 negative patients without CMV reactivation.

**Table 5. Variables associated with mortality at day 60 in patients with CMV reactivation only (n = 134). Unadjusted and adjusted analysis.**

| | Unadjusted | | | Adjusted | | |
|---|---|---|---|---|---|---|
| | HR | 95% CI | P-value | OR | 95% CI | P-value |
| SARS-CoV-2 status | 1.07 | 0.62-1.84 | 0.81 | 1.80 | 0.91-3.59 | 0.09 |
| Antiviral Treatment | 0.57 | 0.33-0.97 | 0.04 | 0.40 | 0.22-0.74 | <0.01 |
| Acute respiratory failure | 1.93 | 0.87-4.27 | 0.10 | 2.84 | 1.08-7.45 | 0.03 |
| Hematologic malignancy | 2.66 | 0.95-7.47 | 0.06 | 5.79 | 1.66-20.24 | 0.01 |
| Norepinephrine use | 0.119 | 0.03-0.52 | <0.01 | 0.14 | 0.03-0.64 | 0.01 |
| Age | 1.06 | 1.03-1.09 | <0.001 | 1.05 | 1.02-1.08 | <0.001 |
| SAPS 2 | 1.02 | 1.00-1.04 | 0.03 | 1.04 | 1.01-1.06 | <0.01 |
| Time from hospital admission to intubation | 1.02 | 0.99-1.04 | 0.07 | 1.03 | 1.01-1.05 | 0.02 |

HR=Hazard Ratio; OR=Odds Ration; CI=Confidence Interval.

reactivation. This ponders the question whether blood or lung CMV reactivation is more relevant when trying to predict a potential clinical impact of this viral reactivation.

Evaluating whether CMV reactivation is more common in SARS-CoV-2 positive patients requires comparing such a population to a control population. We chose to compare patients who required at least four days of IMV regardless of the indication as this was done in a randomized controlled trial evaluating the preemptive treatment of CMV [14]. One of the strengths of our study rests in the fact that we compared SARS-CoV-2 negative patients treated during the same period as SARS-CoV-2 positive patients therefore limiting the risk of bias due to changes in standard of care.

Whereas this choice reduces the homogeneity of the control population reducing internal validity, it increases external validity. To this date, only Luyt et al. proposed a comparative study focusing on the incidence of CMV lung reactivation in two cohorts of patients suffering from severe ARDS related to either influenza virus or SARS-CoV-2 [13]. After considering competing factors, no significant difference was found between the two cohorts (p = 0.07) matching our results.

Our work showed a very strong association between administration of methylprednisolone and CMV blood and/or lung reactivation regardless of SARS-CoV-2 status. Conversely, dexamethasone was not associated with CMV reactivation. Luyt et al. [13] found no association between steroids and CMV reactivation in severe Coronavirus-Disease-2019 (COVID-19) patients. Here, we aimed to differentiate between the early use of

dexamethasone (used as the standard of care for SARS-CoV-2 pneumonia) and the later use of methylprednisolone (for persistent ARDS). It appeared that methylprednisolone, mainly from the second week of IMV or even later to treat patients with ARDS related lung fibrosis (as described by Meduri et al. [22]) favored CMV reactivation. This could be particularly true when methylprednisolone is used for a long time, especially when started after prolonged IMV. The interest of methylprednisolone during ARDS is highly debated. Its late administration could be associated with poor outcome [23] because of increasing the risk of secondary infections (bacterial or viral) in the context of "ICU related" immunodepression.

We found a significant increase in 60-day mortality in SARS-CoV-2 positive patients with CMV blood or lung reactivation compared to others. Interestingly, there was no difference in 60-day mortality between SARS-CoV-2 positive patients without CMV reactivation, SARS-CoV-2 negative patients with CMV reactivation and SARS-CoV-2 negative patients without CMV reactivation when comparing them two by two. Whereas CMV blood or lung reactivation may be the cause of increased mortality in ICU patients is still matter of debate. However, CMV blood reactivation in non-immunocompromised intensive care patients has been found in numerous studies to be associated with poorer outcome [1,24]. This raises the question of the mechanism behind the poorer outcome found in patients presenting a CMV and SARS-CoV-2 co-infection.

Roughly 50% of the patients presenting CMV reactivation received a curative dose of anti-viral treatment by ganciclovir. Among patients with CMV reactivation, anti-viral treatment was found to be a protective factor regarding 60-day mortality. Administration of antiviral treatment was decided following the algorithm proposed by Papazian et al [25]. Whether *Herpsviridae* (CMV, Herpes simplex virus but also Epstein-Baar virus) reactivations among SARS-CoV-2 critically-ill patients represent a cause of hyperinflammation and deserve targeted treatment remains uncertain [26]. As our study was not designed to evaluate the treatment of CMV reactivation, no formal conclusion can be drawn from this result. However, it may encourage the design of future trials evaluating the treatment of CMV infection in mechanically ventilated critical patients.

Our study has several limitations, the first being its retrospective monocenter design exposing our results to a selection bias. Secondly, CMV serological status was available for only half patients. However, they were similarly distributed in SARS-CoV-2 positive and negative groups. In patients for which serological status was known, the rate of seropositivity was high, consistently with what is usually described among ICU adult patients [27] and similar in SARS-CoV-2 positive and negative groups. These elements advocate for the validity of the analyses performed in our whole cohort. Complementary analyses on seropositive patients only confirmed the role of methylprednisolone as a risk factor for CMV reactivation in both SARS-CoV-2 positive and negative patients. We chose to define CMV reactivation as either blood or/and lung but not all patients beneficiated from lung and blood CMV tests, therefore, no conclusion can be drawn on whether lung reactivation or blood reactivation is more related to a clinical impact.

Even though the majority of patients in both groups were admitted for ARDS, SARS-CoV-2 positive and negative patients differed in many ways especially on cancer, hematologic malignancies and immunosuppressive treatments thus limiting internal validity of comparisons. Our statistical analysis was censored after 60 days post intubation, even though only few patients were censored (roughly 2%). Our results are only applicable in this time range.

## Conclusion

In this large retrospective cohort, CMV reactivation was not significantly associated with SARS-CoV-2 infection in patients undergoing invasive mechanical ventilation for at least 4 days. The major risk factor for CMV reactivation was treatment with methylprednisolone whether in patients with SARS-CoV-2 infection or not. The combination of CMV reactivation in patients under invasive mechanical ventilation with SARS-CoV-2 infection was associated with higher mortality but ganciclovir treatment had a protective effect. How SARS-CoV-2 and CMV interact resulting in a poorer outcome deserves to be more precisely investigated.

## Supporting information

**S1 Table. Factors associated with CMV reactivation in the sub-population with known seropositive CMV status (n = 129).** Stepwise downward selection method forcing the variable « SARS-CoV-2 positive ».
(DOCX)

**S2 Table. Risk factors for mortality in the first 60 days post intubation in the sub-population with known seropositive CMV status (n = 129).** Adjusted and unadjusted analysis.
(DOCX)

## Author contributions

**Conceptualization:** Octave Cannac, Xavier de Lamballerie, Rémi Charrel, Laurent Papazian, Sami HRAIECH.

**Investigation:** Octave Cannac, Léa Luciani, Paul-Rémi Petit, Sami HRAIECH.

**Methodology:** Octave Cannac, Vanessa Pauly, Damien Barrau, Geoffray Agard, Sami HRAIECH.

**Supervision:** Vanessa Pauly, Christine Zandotti, Xavier de Lamballerie, Rémi Charrel, Laurent Papazian, Sami HRAIECH.

**Validation:** Christine Zandotti, Sami HRAIECH.

**Writing – original draft:** Octave Cannac, Vanessa Pauly, Christine Zandotti, Léa Luciani, Paul-Rémi Petit, Xavier de Lamballerie, Rémi Charrel, Damien Barrau, Geoffray Agard, Sami HRAIECH.

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
