## [Decision Letter · Decision Letter 0]

31 Jul 2025

Dear Dr. HRAIECH,

Thank you for submitting your manuscript to PLOS ONE. After careful consideration, we feel that it has merit but does not fully meet PLOS ONE’s publication criteria as it currently stands. Therefore, we invite you to submit a revised version of the manuscript that addresses the points raised during the review process.

Please clarify CMV testing criteria and rationale, improve methodological transparency, address the interpretation of controversial results, and place findings in the context of existing literature.

We look forward to receiving your revised manuscript.

Kind regards,

Glenda Canderan, PhD

Academic Editor

PLOS ONE

Journal Requirements:

3. We note that your Data Availability Statement is currently as follows: All relevant data are within the manuscript and its Supporting Information files

Reviewers' comments:

Reviewer's Responses to Questions

**Comments to the Author**

1. Is the manuscript technically sound, and do the data support the conclusions?

Reviewer #1: Yes

Reviewer #2: Partly

Reviewer #3: Partly

2. Has the statistical analysis been performed appropriately and rigorously?

Reviewer #1: Yes

Reviewer #2: I Don't Know

Reviewer #3: No

3. Have the authors made all data underlying the findings in their manuscript fully available?

Reviewer #1: Yes

Reviewer #2: No

Reviewer #3: Yes

4. Is the manuscript presented in an intelligible fashion and written in standard English?

Reviewer #1: Yes

Reviewer #2: Yes

Reviewer #3: Yes

Reviewer #1: The manuscript by Cannac et al. assesses the correlation between reactivation of cytomegalovirus (CMV) in SARS-CoV-2-infected patients versus non-SARS-CoV-2-infected patients and the effect on patient outcome. The manuscript is well-written and informative. There are a few suggested changes that would improve the manuscript.

Major comment:

1) Several previous studies have analyzed CMV reactivation in SARS-CoV-2 patients with mechanical ventilation; it is not clearly stated in the manuscript how the current study adds to or differs from these studies. There should be more information about this in the introduction and especially the discussion.

Minor comment:

1) Lines 54-56 is a bit confusing and should be restated.

Reviewer #2: Cytomegalovirus reactivation in mechanically ventilated patients with or without SARS-CoV-2 infection: A comparative study

This manuscript presents a monocentric retrospective cohort study comparing the incidence, risk factors, and outcomes of cytomegalovirus (CMV) reactivation in mechanically ventilated patients with and without SARS-CoV-2 infection. The study includes 330 patients, the authors used competing risk analysis (Fine & Gray model) and Cox regression to estimate the probability of CMV reactivation.

The authors report an association of treatment with methylprednisolone but not of SARS-CoV-2 infection with CMV reactivation . Moreover, among patients presenting a CMV reactivation, the administration of ganciclovir was associated with lower mortality.

Comments:

• The authors state, that all data are reported to be available without restriction, however, the submitted manuscript does not contain primary data.

• Overall, the retrospective nature of the study limits causal inference and increases the risk of selection and information bias. Although this limitation is somewhat acknowledged in the discussion, the authors draw unsupported conclusions from the data presented. The indication for CMV testing remains somewhat dubious (routine in patients after 4 days MV? Why?) and I would suspect that the test was ordered in patients not responding to treatment in timely manner, implying an important selection of patients with higher mortality. Was any attempt made to increase comparability of the two cohorts, e.g. propensity score matching accounting for known prognostic factors?

• Why was the cutoff of four days chosen? Is there any sound rationale for excluding patients based on the cutoff?

• In accordance to this potential bias, the SARS-CoV-2-negative group is heterogeneous in terms of admission diagnoses, which may affect the internal validity of comparisons. The authors acknowledge this limitation, but it should be more explicitly discussed in the Discussion section.

• CMV serostatus data are missing for more than half of the patients, and there was no standardized protocol for CMV testing. This may lead to underestimation of CMV reactivation incidence and limits the generalizability of results to populations with different CMV seroprevalence.

• The authors report, that methylprednisolone was strongly associated with CMV reactivation (SHR 2.84, p<0.001), regardless of SARS-CoV-2 status but dexamethasone was not associated with CMV reactivation. This seems counterintuitive and warrants further explanation.

• Antiviral Treatment: The possible protective effect of ganciclovir on mortality is of interest but should be interpreted with caution due to the non-randomized, retrospective nature of the data. The authors appropriately note that this finding should be considered hypothesis-generating for future studies. Also, suggest to cite Naendrup JH, Garcia Borrega J, Eichenauer DA, Shimabukuro-Vornhagen A, Kochanek M, Böll B. Reactivation of EBV and CMV in Severe COVID-19-Epiphenomena or Trigger of Hyperinflammation in Need of Treatment? A Large Case Series of Critically ill Patients. J Intensive Care Med. 2022 Sep;37(9):1152-1158. doi: 10.1177/08850666211053990. Epub 2021 Nov 18. PMID: 34791940; PMCID: PMC9396115.

Reviewer #3: The study analyzes CMV reactivation in patients in a tertiary center, evaluating the risk of reactivation and the factor of exposure to SARS-CoV-2 infection. The methodology included survival analysis with competing risks to assess the risk of CMV reactivation across groups. This analysis is, in fact, recommended. However, a few points remain to be addressed to indicate that it was carried out properly. Furthermore, some uncertainty levels are still missing, especially in Figures 2 and 3. Also, a few issues deserve attention or need clarification.

Most importantly, it is unclear how the information about patient status regarding CMV exposure prior to hospitalization was obtained. If a patient had a CMV reactivation, I assume it means that this patient had been exposed to CMV prior to hospitalization. However, it should be clear that this comparison involves how patients were assessed for CMV exposure, with reactivation considered only for those with previous exposure.

Additionally, the methodology should clearly outline how CMV activation was treated in the analysis, particularly for mortality, which is said to be time-dependent.

The title suggests a comparative study. However, there is no distinct class of studies named "comparative study", and many studies inherently involve some form of comparison. My point is that "comparative study" does not convey the type of study in a title. I suggest rethinking the title.

The manuscript is well-written, however, it would benefit from some revision on language and style. For instance, in lines 152-153: "so as to distinguish between clinical groups and so to be more informative"

Abstract: HR=0,4 (use point for decimals). Revise the manuscript for this kind of mistake.

Abstract- Measurements (lines 34-36): The abstract does not mention the use of survival analysis, an important component of the methodology. I recommend including it, even though it might require rewriting the text due to its length. Also, it does not mention primary and secondary endpoints.

Background (or Introduction section): incidence ranging from 8% to 46% (line 54). Usually, the term incidence has a time component, such as measured over person-years. In this case, it appears to be a proportion of the events, such as the ratio between the number of people with reactivation and the total population of ICU patients or the population that had had CMV in the past. It seems the manuscript refers to the former, but (1) it has to be clear, and (2) this could be the reason for the wide range and should be discussed, although not in this section.

Background: It appears from lines 69-70 (Refs 10-12) that reactivation has been measured in previous studies. If reactivation has indeed been measured, explain why the statement in 74-75 remains true. In other words, explain what the novelty is in this study.

Methodology:

How was CMV activation treated as a time-dependent variable in the mortality analysis? I recommend making it clear.

Line 147: Please clarify what it means that the SARS-CoV-2 positive variable was forced into the analysis.

Some colors used in Figures 2 and 3 are difficult to distinguish. I recommend using a more distinguishable set of colors.

In Figures 2 and 3, the uncertainty levels should be included.

For Figure 2, how was probability computed? It seems that it was the proportion of individuals with the signaled events over time. If so, I would rather say proportion of frequency instead of probability.

For Figure 2, the caption defines T as time in days. I strongly recommend changing the label of the x-axis to express this information.

Other minor issues:

Line 85: correct "all patients form…".

Line 128: "if not the" - wording sounds strange.

**Do you want your identity to be public for this peer review?** For information about this choice, including consent withdrawal, please see our Privacy Policy

Reviewer #1: No

Reviewer #2: No

Reviewer #3: No

---

## [Author Response · Author response to Decision Letter 1]

8 Aug 2025

PONE-D-25-24190

Cytomegalovirus reactivation in mechanically ventilated patients with or without SARS-CoV-2 infection: A comparative study.

PLOS ONE

Dear Dr. HRAIECH,

Thank you for submitting your manuscript to PLOS ONE. After careful consideration, we feel that it has merit but does not fully meet PLOS ONE’s publication criteria as it currently stands. Therefore, we invite you to submit a revised version of the manuscript that addresses the points raised during the review process.

Please clarify CMV testing criteria and rationale, improve methodological transparency, address the interpretation of controversial results, and place findings in the context of existing literature.

We look forward to receiving your revised manuscript.

Kind regards,

Glenda Canderan, PhD

Academic Editor

PLOS ONE

Journal Requirements:

3. We note that your Data Availability Statement is currently as follows: All relevant data are within the manuscript and its Supporting Information files

Reviewers' comments:

Reviewer's Responses to Questions

Comments to the Author

1. Is the manuscript technically sound, and do the data support the conclusions?

Reviewer #1: Yes

Reviewer #2: Partly

Reviewer #3: Partly

2. Has the statistical analysis been performed appropriately and rigorously?

Reviewer #1: Yes

Reviewer #2: I Don't Know

Reviewer #3: No

3. Have the authors made all data underlying the findings in their manuscript fully available?

Reviewer #1: Yes

Reviewer #2: No

Reviewer #3: Yes

4. Is the manuscript presented in an intelligible fashion and written in standard English?

Reviewer #1: Yes

Reviewer #2: Yes

Reviewer #3: Yes

5. Review Comments to the Author

Reviewer #1:

The manuscript by Cannac et al. assesses the correlation between reactivation of cytomegalovirus (CMV) in SARS-CoV-2-infected patients versus non-SARS-CoV-2-infected patients and the effect on patient outcome. The manuscript is well-written and informative. There are a few suggested changes that would improve the manuscript.

Major comment:

1) Reviewer’s comment:

Several previous studies have analyzed CMV reactivation in SARS-CoV-2 patients with mechanical ventilation; it is not clearly stated in the manuscript how the current study adds to or differs from these studies. There should be more information about this in the introduction and especially the discussion.

Authors’ response: we agree with the reviewer’s comment. We performed a careful re-analysis of the literature data available in this field. Several studies have limited effectives (under 100 patients) and their results, both on the incidence of CMV reactivation in SARS-CoV-2 patients and its impact on mortality are conflicting. Using competing risk methods, we did not confirm an association between SARS-CoV2 and CMV but we showed a negative effect on mortality of viral co-infection. The main originality of our cohort, apart from its size, stays in the fact that we put in light the role of methylprednisolone (and not dexamethasone) as a risk factor for CMV reactivation and a lower mortality in patients that were treated with ganciclovir. We placed our findings in the context of existing literature and what our study adds to the field in the introduction but especially in the discussion sections.

Minor comment:

1) Reviewer’s comment:

Lines 54-56 is a bit confusing and should be restated.

Authors ‘response:

This sentence has been reworded.

We’d like to thank the reviewer for his helpful comments

Reviewer #2:

Cytomegalovirus reactivation in mechanically ventilated patients with or without SARS-CoV-2 infection: A comparative study

This manuscript presents a monocentric retrospective cohort study comparing the incidence, risk factors, and outcomes of cytomegalovirus (CMV) reactivation in mechanically ventilated patients with and without SARS-CoV-2 infection. The study includes 330 patients, the authors used competing risk analysis (Fine & Gray model) and Cox regression to estimate the probability of CMV reactivation.

The authors report an association of treatment with methylprednisolone but not of SARS-CoV-2 infection with CMV reactivation . Moreover, among patients presenting a CMV reactivation, the administration of ganciclovir was associated with lower mortality.

Reviewer’s comment:

• The authors state, that all data are reported to be available without restriction, however, the submitted manuscript does not contain primary data.

Authors ‘response:

Primary data have been uploaded as Supporting Information files.

Reviewer’s comment:

• Overall, the retrospective nature of the study limits causal inference and increases the risk of selection and information bias. Although this limitation is somewhat acknowledged in the discussion, the authors draw unsupported conclusions from the data presented.

Authors ‘response:

We agree with the reviewer that the retrospective design of the study is a weakness and have stated it as the first limit as noted by the reviewer. Considering this, we have described our results in a careful manner and only presented the conclusions supported by strong statistical methods.

Reviewer’s comment:

The indication for CMV testing remains somewhat dubious (routine in patients after 4 days MV? Why?) and I would suspect that the test was ordered in patients not responding to treatment in timely manner, implying an important selection of patients with higher mortality.

Authors ‘response:

We acknowledge that the routine diagnosis of CMV reactivation in ICU patients is not consensual. Moreover, our description of CMV screening was confusing as we declared that there was no protocol for CMV screening. This statement concerned only respiratory tract CMV screening but not blood screening. We apologize for this omission and imprecision in the methods section.

In fact, during the study period, patients under IMV for at least 4 days were supposed to be screened twice weekly for CMV reactivation with PCR on whole blood for as long as they remained under IMV. This has become part of routine care in our ICU since a randomized controlled trial investigating the effect of preemptive ganciclovir in ICU patients with CMV reactivation (doi.org/10.1186/s13613-020-00793-2). In this trial, patients had a routine screening of CMV reactivation from the 4th day of MV. This delay was determined from previous studies showing that CMV reactivation before this time point was rarely described and that CMV reactivation more frequently occurred after this time point (see response lower). Since this trial, we kept on practicing CMV routine screening in patients undergoing at least 4 days of MV as part of routine care, even though, as for all protocols of care, the observance was not perfect. This screening is therefore not correlated with patients’ severity.

Conversely, there was no protocol for CMV screening in LRT which was decided by clinicians according to patient’s evolution.

This has been added in the methods section (subheading “CMV reactivation and SARS-CoV-2 positivity”)

Reviewer’s comment:

Was any attempt made to increase comparability of the two cohorts, e.g. propensity score matching accounting for known prognostic factors?

Authors ‘response:

We increased comparability between the cohorts by adjusting for key prognostic variables in multivariable models. While propensity score methods are valuable, we chose direct covariate adjustment to preserve sample size and transparency.

Indeed, we didn’t want to decrease our statistical power by loosing patients during the matching process. Moreover, we also wanted to analyze explicitly the effect of the other factors on CMV reactivation and mortality, for example treatment with methylprednisolone.

We agree that propensity score methods (e.g., matching, weighting) are valuable tools to improve comparability between groups. Here, we prioritized covariate adjustment because the number of events allowed for multivariate modeling without risk of overfitting. However, we acknowledge that propensity score techniques could be considered in future sensitivity analyses.

Reviewer’s comment:

• Why was the cutoff of four days chosen? Is there any sound rationale for excluding patients based on the cutoff?

Authors ‘response:

As previously mentioned, this was the inclusion criteria in the “Premptive treatment for herpesviridae” study (doi.org/10.1186/s13613-020-00793-2) since then applied as a part of routine care in our ICU. The rationale supporting this 4-day delay is based on last decade’s trials investigating CMV reactivation among ICU patients. These trials showed that CMV disease typically occurs within the first 2 weeks of critical illness (reference: Al‑Omari et al. Ann. Intensive Care (2016) 6:110). Notably, the reported incidence of the disease was much higher in studies that screened the patients weekly than those that screened only once within the first 4 days of admission to the ICU, indicating that the disease happens frequently beyond the first 4 days post-admission. Among the studies using PCR detection, the mean and median times of detection ranged from 4 to 12 days after ICU admission, especially in patients undergoing MV which is a known risk factor for CMV reactivation among immunocompetent ICU patients. (Osawa R, Singh N. Cytomegalovirus infection in critically ill patients: a systematic review. Crit Care. 2009;13(3):R68.).

Reviewer’s comment:

• In accordance to this potential bias, the SARS-CoV-2-negative group is heterogeneous in terms of admission diagnoses, which may affect the internal validity of comparisons. The authors acknowledge this limitation, but it should be more explicitly discussed in the Discussion section.

Authors’ response:

We agree with the reviewer remark that IMV indication in SARS-CoV-2 negative patients varied. However, this heterogeneity increases external validity. Moreover, the majority of patients in b

---

## [Decision Letter · Decision Letter 1]

29 Aug 2025

Dear Dr. HRAIECH,

We look forward to receiving your revised manuscript.

Kind regards,

Glenda Canderan, PhD

Academic Editor

PLOS ONE

Journal Requirements:

**Additional Editor Comments:**

The manuscript has undergone significant improvement, and two of the three reviewers are now satisfied with the changes. One reviewer, however, has raised a remaining point that I believe should be addressed to strengthen the work.

Specifically, the reviewer notes that information about CMV status at admission was only available for half of the patients. Given that the manuscript focuses on CMV reactivation, the reviewer recommends that you provide additional consideration of this limitation. While I understand it may not be possible to resolve fully due to the retrospective nature of the study, I ask that you please:

**Clarify and expand the discussion**  regarding the potential biases introduced by missing CMV serostatus at admission. **Revise Table 1 (if feasible)**  to include the number of patients with known CMV status, stratified by SARS-CoV-2 positive and negative groups.

 Moreover:

**Revisit the discussion of corticosteroid use (methylprednisolone vs. dexamethasone):**  this point comes across more clearly in your response to reviewers than in the main text.**Address language/formatting issues:**  please carefully proofread the newly added text to correct typos,  improve clarity and remove redundancies.

 I do not anticipate that additional analyses will be required if a careful discussion is provided; however, if you are able to perform a sensitivity analysis restricted to patients with known CMV status, that would further strengthen the manuscript.

Please submit a revised version with a short response letter indicating how you have addressed these points.

Reviewers' comments:

Reviewer's Responses to Questions

**Comments to the Author**

Reviewer #1: All comments have been addressed

Reviewer #2: All comments have been addressed

Reviewer #3: (No Response)

2. Is the manuscript technically sound, and do the data support the conclusions?

Reviewer #1: Yes

Reviewer #2: Yes

Reviewer #3: Partly

3. Has the statistical analysis been performed appropriately and rigorously?

Reviewer #1: Yes

Reviewer #2: Yes

Reviewer #3: No

4. Have the authors made all data underlying the findings in their manuscript fully available?

Reviewer #1: Yes

Reviewer #2: Yes

Reviewer #3: Yes

5. Is the manuscript presented in an intelligible fashion and written in standard English?

Reviewer #1: Yes

Reviewer #2: Yes

Reviewer #3: Yes

Reviewer #1: The authors should be commended for their nice study and thorough responses to the reviewers. I have no additional concerns.

Reviewer #2: Please exclude patient IDs from the data as standard recomendations in order to protect patinets. Otherwise very good revision, thank you!

Reviewer #3: The manuscript underwent significant improvement - however, in light of the fact that information about CMV status at admission was available for only half of the patients, I believe that more should be described about CMV status at admission. For instance, Table 1 can include n of CMV status positive and CMV status for SARS-CoV-2 positive and negative patients.

Since CMV status was not part of the inclusion/exclusion criteria, the study had alternatives for the survival analysis: (1) include only individuals with CMV status - in this case, losing statistical power and incurring a bias failing to consider several potential CMV exposed individuals; and (2) include all individuals - in this case many individuals would not experience reactivation because were not exposed, which is a different bias.

I recommend evaluating the results with additional analysis using a subset that had CMV status at admission and comparing the effects in both analyses. Then, a necessary discussion will be warranted with these options and the results, rather than stating that CMV serological status was missing for half the population and therefore not shown. In fact, the manuscript is about CMV reactivation, and this evaluation is important to comprehend the potential biases.

Line 354-355 "freeing us from the bias of changing standards of care"- wording sounds strange, I recommend revising it.

Choice of colors in Figures: I do think it was possible to change the colors in Figures; but this was an optional recommendation.

Line : "CMV serological status was missing for more than half of our population which is why we chose not to display these

data" - I do not agree. It is actually one more reason to present this data, as this variable is important.

Correct "Student T test": Student's t-test.

**Do you want your identity to be public for this peer review?** For information about this choice, including consent withdrawal, please see our Privacy Policy

Reviewer #1: No

Reviewer #2: No

Reviewer #3: No

---

## [Author Response · Author response to Decision Letter 2]

17 Sep 2025

Dear Editor and Reviewers,

Thank you for your comments and for offering us the possibility to submit a revised version of our manuscript.

Please find enclosed our responses to the points you raised.

Additional Editor Comments:

Editor’s comment:

The manuscript has undergone significant improvement, and two of the three reviewers are now satisfied with the changes.

Authors’ response:

We thank the Editor for this positive appreciation.

Editor’s comment:

One reviewer, however, has raised a remaining point that I believe should be addressed to strengthen the work.

Specifically, the reviewer notes that information about CMV status at admission was only available for half of the patients. Given that the manuscript focuses on CMV reactivation, the reviewer recommends that you provide additional consideration of this limitation. While I understand it may not be possible to resolve fully due to the retrospective nature of the study, I ask that you please:

1. Clarify and expand the discussion regarding the potential biases introduced by missing CMV serostatus at admission.

Author’s response:

We understand the importance of such a question. Therefore we decided to perform a complementary analysis to evaluate if the lack of serological status for about half of our population might have represented an important bias in our results.

CMV serological status at ICU admission was available for 166 (50.3%) patients. There was no difference between the two groups: 75/147 (51.0%) of SARS-CoV-2 negative patients had a known CMV serological status vs. 91/183 (49.7%) for the SARS-CoV-2 positive patients (p=0.81). Among patients with a known CMV serological status, 129 (77.8%) were seropositive. This result is in line with what is usually described: 60-80% of adults are CMV seropositive in western countries and it has been shown that seroprevalence gradually increases from 50% in young adults to 90% in the elderly (Papazian et al. Intensive Care Med (2016) 42:28–37 DOI 10.1007/s00134-015-4066-9; Ong et al. DOI: 10.1097/CCM.0000000000000712). Considering the mean age of our population, such a seroprevalence is not surprising. When focusing on the patients for which the serological status was known, 71 (78%) SARS-CoV-2 positive patients were seropositive and 58 (77%) SARS CoV-2 negative patients were seropositive (p=0.95). We think that the equivalent repartition of seropositive patients among COVID-19 and non-COVID-19 patients is rather reassuring towards the risk of bias. These data have been added in the methods section and in the results section, in the text and in the table 1.

To confirm our results, we also performed sensitivity analyses focusing only on patients for which serological status was known and who were seropositive, therefore that could effectively reactivate CMV during their ICU stay. The results of these complementary analyses have been added in the supplementary material (Tables S1 and S2). Concerning CMV reactivation, methylprednisolone use remained a significant risk factor in this sub-population (129 patients). Time to intubation did not reach significance anymore (p=0.06) probably because of a lack of statistical power. Concerning mortality risk factors, in this sub-population, SARS-CoV-2 and CMV reactivation did not reach significance any more (p= 0.06 and 0.09 respectively) probably because of a loss of statistical power.

Lastly, the discussion has been modified according to these elements. Serological status is now clearly addressed in the discussion section in which we acknowledge the limits of missing data and bring in light the elements that comfort our results despite this limitation.

Overall, considering the equivalent repartition of CMV known status and CMV seropositivity in both groups with a high prevalence of CMV seropositivity, we remain confident about the results in our whole cohort.

Editor’s comment:

Revise Table 1 (if feasible) to include the number of patients with known CMV status, stratified by SARS-CoV-2 positive and negative groups.

Authors’response:

This has been done as requested.

Editor’s comment:

Moreover:

• Revisit the discussion of corticosteroid use (methylprednisolone vs. dexamethasone): this point comes across more clearly in your response to reviewers than in the main text.

Authors’ response:

Following the Editor advice, we have clarified and developed this point of the discussion concerning the effects of methylprednisolone but not dexamethasone on CMV reactivation putting it in light the still ongoing debate on benefits and risks of steroids in ARDS patients.

Editor’s comment:

• Address language/formatting issues: please carefully proofread the newly added text to correct typos, improve clarity and remove redundancies.

Authors’ response:

The manuscript has been carefully reread in this sense.

Editor’s comment:

I do not anticipate that additional analyses will be required if a careful discussion is provided; however, if you are able to perform a sensitivity analysis restricted to patients with known CMV status, that would further strengthen the manuscript.

Authors’ response:

These sensitivity analyses have been done and added in the supplementary material.

Please submit a revised version with a short response letter indicating how you have addressed these points.

Reviewers' comments:

Reviewer's Responses to Questions

Comments to the Author

1. If the authors have adequately addressed your comments raised in a previous round of review and you feel that this manuscript is now acceptable for publication, you may indicate that here to bypass the “Comments to the Author” section, enter your conflict of interest statement in the “Confidential to Editor” section, and submit your "Accept" recommendation.

Reviewer #1: All comments have been addressed

Reviewer #2: All comments have been addressed

Reviewer #3: (No Response)

2. Is the manuscript technically sound, and do the data support the conclusions?

Reviewer #1: Yes

Reviewer #2: Yes

Reviewer #3: Partly

3. Has the statistical analysis been performed appropriately and rigorously?

Reviewer #1: Yes

Reviewer #2: Yes

Reviewer #3: No

4. Have the authors made all data underlying the findings in their manuscript fully available?

Reviewer #1: Yes

Reviewer #2: Yes

Reviewer #3: Yes

5. Is the manuscript presented in an intelligible fashion and written in standard English?

Reviewer #1: Yes

Reviewer #2: Yes

Reviewer #3: Yes

6. Review Comments to the Author

Reviewer #1:

Reviewer’s comment:

The authors should be commended for their nice study and thorough responses to the reviewers. I have no additional concerns.

Authors’response:

We thank the reviewer for his/her comment and positive appreciation of our work.

Reviewer #2:

Reviewer’s comment:

Please exclude patient IDs from the data as standard recomendations in order to protect patients. Otherwise very good revision, thank you!

Authors’ response:

Patients’ identification numbers have been deleted from the primary data. We thank the reviewer for his/her comment and positive appreciation of our work.

Reviewer #3:

Reviewer’s comment:

The manuscript underwent significant improvement - however, in light of the fact that information about CMV status at admission was available for only half of the patients, I believe that more should be described about CMV status at admission. For instance, Table 1 can include n of CMV status positive and CMV status for SARS-CoV-2 positive and negative patients.

Authors’ response:

We agree with the reviewer about the importance of CMV status when evaluating CMV reactivation. Therefore, we analyzed the subpopulation for whom the serological status for CMV was known at ICU admission.

CMV serological status at ICU admission was available for 166 (50.3%) patients. There was no difference between the two groups: 75/147 (51.0%) of SARS-CoV-2 negative patients had a known CMV serological status vs. 91/183 (49.7%) for the SARS-CoV-2 positive patients (p=0.81). Among patients with a known CMV serological status, 129 (77.8%) were seropositive. This result is in line with what is usually described: 60-80% of adults are CMV seropositive in western countries and it has been shown that seroprevalence gradually increases from 50% in young adults to 90% in the elderly (Papazian et al. Intensive Care Med (2016) 42:28–37 DOI 10.1007/s00134-015-4066-9; Ong et al. DOI: 10.1097/CCM.0000000000000712). Considering the mean age of our population, such a seroprevalence is not surprising. When focusing on the patients for which the serological status was known, 71 (78%) SARS-CoV-2 positive patients were seropositive and 58 (77%) SARS CoV-2 negative patients were seropositive (p=0.95). We think that the equivalent repartition of seropositive patients among COVID-19 and non-COVID-19 patients is rather reassuring towards the risk of bias. These data have been added in the methods section and in the results section, in the text and in the table 1.

Reviewer’s comment:

Since CMV status was not part of the inclusion/exclusion criteria, the study had alternatives for the survival analysis: (1) include only individuals with CMV status - in this case, losing statistical power and incurring a bias failing to consider several potential CMV exposed individuals; and (2) include all individuals - in this case many individuals would not experience reactivation because were not exposed, which is a different bias.

I recommend evaluating the results with additional analysis using a subset that had CMV status at admission and comparing the effects in both analyses.

Authors’ response:

According to the reviewer’s recommendation, to confirm our results, we performed sensitivity analyses focusing only on patients for which serological status was known and who were seropositive (reactivation among seronegative patients during the ICU stay being highly unlikely), therefore that could effectively reactive CMV during their ICU stay.

The results of these complementary analyses have been added in the supplementary material (Tables S1 and S2). Concerning CMV reactivation, methylprednisolone use remained a significant risk factor in this sub-population (129 patients). Time to intubation did not reach significance anymore (p=0.06) probably because of a lack of statistical power. Concerning mortality risk factors, in this sub-population, SARS-CoV-2 and CMV reactivation did not reach significance any more (p= 0.06 and 0.09 respectively) probably because of a loss of statistical power.

Considering the equivalent repartition of CMV known status and CMV seropositivity in both groups with a high prevalence of CMV seropositivity, we remain confident about the results in our whole cohort and chose to maintain in the main manuscript.

Reviewer’s comment:

Then, a necessary discussion will be warranted with these options and the results, rather than stating that CMV serological status was missing for half the population and therefore not shown. In fact, the manuscript is about CMV reactivation, and this evaluation is important to comprehend the potential biases.

Authors’response:

We agree. The discussion has been modified according to these elements. Serological status is now clearly addressed in the discussion section in which we acknowledge the limits of missing data and bring in light the elements that comfort our results despite this limitation.

Reviewer’s comment:

Line 354-355 "freeing us from the bias of changing standards of care"- wording sounds strange, I recommend revising it.

Authors’response:

Indeed, this sentence has been reworded.

Reviewer’s comment:

Choice of colors in Figures: I do think it was possible to change the colors in Figures; but this was an optional recommendation.

Authors’ response:

We thank the reviewer for his/her understanding.

Reviewer’s comment:

Line : "CMV serological status was missing for more than half of our population which is why we chose not to display these

data" - I do not agree. It is actually one more reason to present this data, as this variable is important.

Authors’response:

This has been modified as described upward.

Reviewer’s comment:

Correct "Student T test": Student's t-test.

Authors’response :

This has been done.

We thank the reviewer for his/her highly relevant comments.

---

## [Decision Letter · Decision Letter 2]

2 Oct 2025

Cytomegalovirus reactivation in mechanically ventilated patients with or without SARS-CoV-2 infection: a retrospective cohort study.

PONE-D-25-24190R2

Dear Dr. HRAIECH,

We’re pleased to inform you that your manuscript has been judged scientifically suitable for publication and will be formally accepted for publication once it meets all outstanding technical requirements.

Kind regards,

Glenda Canderan, PhD

Academic Editor

PLOS ONE

Reviewers' comments:

Reviewer's Responses to Questions

**Comments to the Author**

Reviewer #3: All comments have been addressed

2. Is the manuscript technically sound, and do the data support the conclusions?

Reviewer #3: (No Response)

3. Has the statistical analysis been performed appropriately and rigorously?

Reviewer #3: (No Response)

4. Have the authors made all data underlying the findings in their manuscript fully available?

Reviewer #3: (No Response)

5. Is the manuscript presented in an intelligible fashion and written in standard English?

Reviewer #3: (No Response)

Reviewer #3: All points have been thoroughly addressed, revised, and clearly incorporated in this revised version.

**Do you want your identity to be public for this peer review?** For information about this choice, including consent withdrawal, please see our Privacy Policy

Reviewer #3: No

---

## [Editor Report · Acceptance letter]

PONE-D-25-24190R2

PLOS ONE

Dear Dr. HRAIECH,

I'm pleased to inform you that your manuscript has been deemed suitable for publication in PLOS ONE. Congratulations! Your manuscript is now being handed over to our production team.

Kind regards,

on behalf of

Dr. Glenda Canderan

Academic Editor

PLOS ONE